# High Carbonated Soft Drink Intake is Associated with Health Risk Behavior and Poor Mental Health among School-Going Adolescents in Six Southeast Asian Countries

**DOI:** 10.3390/ijerph17010132

**Published:** 2019-12-23

**Authors:** Supa Pengpid, Karl Peltzer

**Affiliations:** 1ASEAN Institute for Health Development, Mahidol University, Salaya, Phutthamonthon, Nakhonpathom 73170, Thailand; supaprom@yahoo.com; 2Lifestyle Diseases Research Entity, Faculty of Health Sciences, North-West University, Mafikeng Campus, Mmabatho 2745, South Africa

**Keywords:** soft drink intake, aggressive behavior, substance use, psychological distress, adolescents, Southeast Asia

## Abstract

Carbonated soft drink (CSD) intake has been associated with various risk behaviors in adolescents in high-income countries, but there is lack of evidence of this association in cross-nationally representative samples of school adolescents in low- and middle-income countries. This study aimed to assess the association between CSD intake, health risk behavior, and poor mental health behavior among school-going adolescents in six Southeast Asian countries. Cross-sectional national “Global School-Based Student Health Survey (GSHS)” data from 36173 school-going adolescents from Bangladesh, Indonesia, Laos, Philippines, Thailand, and Timor-Leste were analyzed. Results indicate that across all six Southeast Asian countries, in the past 30 days 23.9% of study participants had consumed no CSD, 38.8% had consumed CSD <once/day, 19.9% once a day and 17.5% ≥ two times/day. In the final adjusted logistic regression model CSD intake was associated with increased odds of having been attacked, having sustained an injury, being in a physical fight, being bullied, school truancy, tobacco use, alcohol use, and lifetime drunkenness. In addition, the consumption of CSD ≥two times/day was associated with increased odds of ever used cannabis and ever used amphetamine. Higher intake of CSD was positively associated with a history of loneliness, anxiety, suicide ideation, suicide planning, and suicide attempts. CSD intake in low- and middle-income countries is associated with several health risk behaviors and poor mental health that are similar to those observed in high-income countries.

## 1. Introduction

Carbonated soft drink (CSD) intake has been linked with excess body weight, medical, and oral health problems [1,2]. Relatively little is known of the association between CSD intake and health compromising behavior [3]. Several studies on CSD intake, mainly in high-income countries, found associations between CSD intake and health compromising behaviors.

In studies of American school-going adolescents, higher CSD consumption increased the odds for physical fighting, aggressive behavior [4,5], and substance use [5,6]. In addition, CSD intake was associated with poor mental health, such as depressive symptoms [4,6,7] and suicidal behavior [4]. In school-going adolescents in China, CSD intake was associated with suicidal behavior [8]. In school students in Korea CSD intake was associated with poorer general and mental health [9]; and in an investigation in middle-school students in Malaysia, CSD intake was associated with injury [10]. In a study of adolescents in 26 high-income countries in Europe, sugar consumption increased the odds for health risk behaviors, including substance use (alcohol and cigarettes), physical fighting, and bullying [3].

For optimal physical and mental development during adolescence a balanced diet is vital, and therefore it is of public health concern to understand the relationship between CSD intake and health risk behavior and poor mental health [3]. There is a lack of cross-national studies on the relationship between CSD intake and health compromising behaviors in low- and middle-income countries. The study aimed to assess associations between CSD intake and health risk behavior and poor mental health among school-going adolescents in six Southeast Asian countries.

## 2. Method

### 2.1. Sample and Procedure

This study is a secondary analysis of cross-sectional national data from six Southeast Asian countries collected in 2014–2015 World Health Organization (WHO) “Global School-based Student Health Survey (GSHS)”. Details about the WHO GSHS methodology and its data can be publicly accessed [11]. Briefly, the GSHS uses a cluster sampling design in two stages (schools were selected by probability to size sampling and random selection of class rooms with students 13 to 17 years old) in order to produce nationally representative samples of school children in middle schools in each study country [11]. All students attending a selected class were eligible to participate, regardless of their age, and completed a self-administered questionnaire on a computer scannable answer sheet in their language under the supervision of trained external survey administrators [11]. Ethics review boards in each country approved the GSHS, and informed consent was obtained from the students, parents, and/or school authorities [11].

### 2.2. Measures

The survey parameters of the GSHS are described in detail in Table 1 [11]. Overweight is “defined as more than +1 standard deviation (SD) from the median body mass index by age and sex, and underweight less than −2SD from median for BMI by age and sex” [11]. Parental or guardian support items were grouped into three levels of support (low = 0–1, medium = 2, and high = 3–4).

### 2.3. Statistical Analysis

Logistic regression analysis was used to estimate the relationship between CSD intake and a number of behavior outcomes separately. In the first model the outcome was adjusted for country and in the second model the outcome was adjusted for country, sex, age, hunger (a proxy for socioeconomic status), body weight, peer support, and parental support. Cases with missing data were excluded. All statistical operations were conducted with “STATA software version 15.0 (Stata Corporation, College Station, TX, USA)”, adjusting for the complex study design.

## 3. Results

### 3.1. Sample Description

The study sample consisted of 36173 school-going adolescents (with a median age of 14 years, interquartile range = 3) from Bangladesh, Indonesia, Laos, Philippines, Thailand, and Timor-Leste; the country response rates ranged from 72% to 94% [11]. Among the six study countries, the lowest number of participants was from Bangladesh (N = 2989, 8.3%) and the highest number from Indonesia (N = 11142, 30.8%). The proportion of female students was 55.1% and male students 44.9%, and 6.6% had mostly or always experienced hunger in the past month. About one in four of the students (26.9%) had been in a physical fight in the past 12 months, 38.6% had been attacked, 38.1% had sustained a serious injury, 29.5% had been bullied, and 26.3% “missed classes or school without permission” in the past month. Regarding substance use, 13.2% of participants were current tobacco users, 10.5% current alcohol users, 10.2% had ever been drunk, 3.6% had ever used cannabis, and 2.8% had ever used amphetamine. In terms of mental health, 10.0% of students reported loneliness, 6.8% anxiety, 7.5% suicide ideation, 8.1% had a suicide plan, and 8.6% had attempted suicide in the past 12 months. Two in five students (40.2%) had almost always or always received peer support in the past month, 52.8% had medium or high parental support, 9.6% were underweight and 13.1% overweight or obese. In all six Southeast Asian countries, in the past 30 days 23.9% of participants had consumed no CSD, 38.8% had consumed CSD <once/day, 19.9% once a day and 17.5% ≥two times/day (see Table 2).

### 3.2. Associations between CSD Intake and Aggression

In the final adjusted logistic regression model, any, once and ≥two times/day CSD intake was associated with increased odds of being attacked, in a physical fight, bullied and injured and of school truancy (see Table 3).

### 3.3. Associations between CSD Intake and Substance Use

In the final adjusted logistic regression model, any, once, and ≥two times/day CSD intake was associated with increased odds of current tobacco use, current alcohol use, and ever been drunk. In addition, ≥two times/day CSD intake was associated with increased odds of ever used cannabis and ever used amphetamine (see Table 4).

### 3.4. Associations between CSD Intake and Poor Mental Health

In the final adjusted logistic regression model, ≥two times/day CSD intake was associated with increased odds of loneliness, anxiety, suicide ideation, suicide plan, and suicide attempt (see Table 5).

## 4. Discussion

The study found that across all six Southeast Asian countries (Bangladesh, Indonesia, Laos, Philippines, Thailand, and Timor-Leste) in 2014–2015, 23.9% of participants had consumed no CSD in the past 30 days, 38.8% had consumed CSD <once/day, 19.9% once a day and 17.5% ≥two times/day. The consumption of CSD appears to be slightly higher than that reported in an earlier study conducted in Malaysia and the Maldives in 2009–2012 [12]. However, the prevalence of ≥once CSD daily intake (37.8%) in this study across six Southeast Asian countries was lower than the corresponding prevalence in 53 low- and middle-income countries (54.3%) [12].

This study found a consistent association between CSD intake and health risk behaviors (being attacked, injury, in a physical fight, being bullied, school truancy, current tobacco use, current alcohol use, ever been drunk, ever used cannabis, and ever used amphetamine) and poor mental health (loneliness, anxiety, suicide ideation, suicide plan, and suicide attempt), independent off relevant confounders. These results are consistent with previous investigations in school-going adolescents in Europe (in terms of health risk behaviors: in a physical fight, smoking and alcohol use) [3], in USA, China, Korea and Malaysia in terms of physical fight, aggressive behavior [4,5], substance use [5,6], and injury [10]), and poor mental health (mental distress or depressive symptoms [4,6,7,9] and suicidal behavior [4,8]). This result that soft drink consumption is accompanied by various health risk behaviors underlines the importance of tackling a clustering of CSD intake with a number of health risk behaviors and poor mental health in health promotion programs in this school population.

The study found that associations between CSD intake and health risk behavior seem to be stronger than with poor mental health. It is possible that adolescents who are more sensation seeking engage more likely in frequent CSD intake and various health risk behaviors [7]. Another, possibility is that because CSD consumption is often initiated earlier (in childhood) [13] than substance use (in adolescence) and peer violence (in adolescence), early soft drink consumption may function as a “gateway” for later substance use [5]. From a problem behavior perspective during adolescence, Jessor and Jessor [14] deposit that problem behaviors, such as CSD intake, substance use and peer violence are interrelated and cluster together.

Carbonated soft drinks contain a high sugar content, and sugar has been found to associated with poor mental health, such as psychological distress and depression [15]. Several possible mechanisms for the relationship between sugar intake and psychological distress and depression have been proposed, e.g., “increased levels of *β*–endorphins and oxidative stress” [15,16]. Moreover, carbonated soft drinks often contain additives, such as caffeine [3]. Caffeine can “potentiate the addictive and toxic effects of drugs of abuse” [17], has been found to increase the risk for in a physical fight among adolescents [4], and was associated with anxiety and depression among school adolescents [18,19].

### Study Limitations

The study was limited by its cross-sectional design of a school population and measures using self-report. Several concepts in this study were assessed with single items, and future investigations may consider including more comprehensive measures. The study did also not assess dietary intake in terms of foods that are high in refined sugar content, such as candy and chocolate, which could be important in relation to sweetened beverage consumption.

## 5. Conclusions

Study results extend previous results from high-income countries showing an association between higher frequency of CSD intake and health risk behaviors (being attacked, injury, in a physical fight, being bullied, school truancy, current tobacco use, current alcohol use, ever been drunk, ever used cannabis, and ever used amphetamines) and poor mental health (loneliness, anxiety, and suicidal behavior), independent off relevant confounders. Frequent CSD intake and its associated health risk behaviors and poor mental health should be targeted in school health promotion interventions.

## Figures and Tables

**Table 1 ijerph-17-00132-t001:** Global school-based Student Health Survey variables, questions, and response options evaluated in this study of 2014–2015 respondents.

Variables	Question	Response Options
Age	“How old are you?”	“11 years old or younger to 18 years old or older”
Sex	“What is your sex?”	“Male, Female”
Hunger	“During the past 30 days, how often did you go hungry because there was not enough food in your home?”	“1 = never to 5 = always (coded 1–3 = 0 and 4–5 = 1)”
Soft drinks	“During the past 30 days, how many times per day did you usually drink carbonated soft drinks, such as… country specific names?”	“1 = not in the past days to 7 = 5 or more times per day (coded 1 = 1, 2 = 2, 3 = 3 and 4–7 = 4)”
	Health risk behavior	
In a physical fight	“During the past 12 months, how many times were you in a physical fight?”	“1 = 0 times to 8 = 12 or more times (coded 1 = 0 and 2–8 = 1)”
Physically attacked	“During the past 12 months, how many times were you physically attacked?”	“1 = 0 times to 8 = 12 or more times (coded 1 = 0 and 2–8 = 1)”
Injury	“During the past 12 months, how many times were you seriously injured?”	“1 = 0 times to 8 = 12 or more times (coded 1 = 0 and 2–8 = 1)”
Bullied	“During the past 30 days, on how many days were you bullied?”	“1 = 0 days to 7 = All 30 days (coded 1 = 0 and 2–7 = 1)”
School truancy	“During the past 30 days, on how many days did you miss classes or school without permission?”	“1 = 0 days to 5 = 10 or more days (coded 1 = 0 and 2–5 = 1)”
Current tobacco use	“During the past 30 days, on how many days did you smoke cigarettes/use any tobacco products other than cigarettes, such as pipes, roll your own cigarettes, or smokeless tobacco?”	“1 = 0 days to 7 = All 30 days (coded 1 = 0 and 2–7 = 1)”
Alcohol use	“During the past 30 days, on how many days did you have at least one drink containing alcohol?”	“1 = 0 days to 7 = All 30 days”
Drunkenness	“During the past 30 days, on the days you drink alcohol, how many drinks did you usually drink per day?”	“1 = Not drink in the past 30 days to 7 = 5 or more drinks (coded 1–3 = 0 and 4–7 = 1)”
Cannabis use	“During your life, how many times have you used marijuana (also called country specific names)?”	“1 = 0 times to 5 = 20 or more times (coded 1 = 0 and 2–5 = 1)”
Amphetamine use	“During your life, how many times have you used amphetamines or methamphetamines (also called country specific names)?”	“1 = 0 times to 5 = 20 or more times (coded 1 = 0 and 2–5 = 1)”
	Mental health	
Loneliness	“During the past 12 months, how often have you felt lonely?”	“1 = never to 5 = always (coded 1–3 = 0 and 4–5 = 1)”
Anxiety	“During the past 12 months, how often have you been so worried about something that you could not sleep at night?”	“1 = never to 5 = always (coded 1–3 = 0 and 4–5 = 1)”
Suicide ideation	“During the past 12 months, did you ever seriously consider attempting suicide?”	“Yes, No”
Suicide plan	“During the past 12 months, did you make a plan about how you would attempt suicide?”	“Yes, No”
Suicide attempt	“During the past 12 months, how many times did you actually attempt suicide?”	“1 = 0 times to 5 = 6 or more times (coded 1 = 0 and 2–5 = 1)”
	Confounding factors	
Peer support	“During the past 30 days, how often were most of the students in your school kind and helpful?”	“1 = never to 5 = always (coded 1–3 = 0 and 4–5 = 1)”
Parental supervision	“During the past 30 days, how often did your parents orguardians check to see if your homework was done?”	“1 = never to 5 = always (coded 1–3 = 0 and 4–5 = 1)”
Parental connectedness	“During the past 30 days, how often did your parents orguardians understand your problems and worries?”	“1 = never to 5 = always (coded 1–3 = 0 and 4–5 = 1)”
Parental bonding	“During the past 30 days, how often did your parents or guardians really know what you were doing with your free time?	“1 = never to 5 = always (coded 1–3 = 0 and 4–5 =1)”
Parental respect for privacy	“During the past 30 days, how often did your parents or guardians go through your things without your approval?”	“1 = never to 5 = always (coded 1–3 = 0 and 4–5 = 1)”
Height	“How tall are you without your shoes on?”	cm
Body weight	“How much do you weigh without your shoes on?”	kg

**Table 2 ijerph-17-00132-t002:** Sample characteristics among adolescents in six Southeast Asian countries.

Variable (#Missing)	Sample	Did Not Drink/Past 30 Days	<1 Time/Day	1 Time/Day	≥2 Times/Day
	N (%)	%	%	%	%
Sociodemographic					
All	36,173	23.9	38.8	19.9	17.5
Country (#0)					
Bangladesh	2989 (8.3)	15.7	37.1	18.6	28.6
Indonesia	11,142 (30.8)	37.6	34.4	18.5	9.4
Laos	3683 (10.2)	12.7	37.1	25.3	25.0
Philippines	8761 (24.2)	12.2	50.6	20.0	17.2
Thailand	5894 (16.3)	11.8	32.0	25.7	30.4
Timor-Leste	3704 (10.2)	13.6	44.5	25.0	16.9
Age (years) (#141)					
≤13	10,327 (34.3)	27.9	34.3	21.0	16.8
14	7919 (26.0)	23.7	37.9	20.2	18.2
15	7348 (19.2)	20.9	40.5	18.4	20.2
≥16	10,438 (20.5)	20.3	45.6	18.8	15.3
Gender (#300)					
Female	19,779 (55.1)	27.0	38.1	19.2	15.7
Male	16,094 (44.9)	20.9	39.5	20.5	19.2
Hunger (#220)	2346 (6.6)	5.3	6.2	7.1	8.2
Health Risk Behavior					
In physical fight (#215)	9208 (26.9)	18.6	27.3	31.0	32.8
Physically attacked (#309)	11,960 (38.6)	30.6	39.9	41.4	42.9
Injury (#5935)	11,510 (38.1)	27.7	38.4	43.1	46.2
Bullied (#2552)	9890 (29.5)	21.0	31.5	32.7	32.9
School truancy (#509)	10,257 (26.3)	19.2	25.2	27.1	37.2
Current tobacco use (#477)	4812 (13.2)	9.1	13.2	13.5	18.1
Current alcohol use (#1027)	4909 (10.5)	4.9	10.8	12.1	16.0
Ever drunkenness (#1170)	4349 (10.2)	5.0	10.6	11.6	14.9
Ever cannabis use (#998)	1296 (3.6)	2.5	3.5	3.9	5.1
Ever amphetamine use (#1088)	985 (2.8)	2.1	2.7	2.9	3.9
Poor Mental Health					
Loneliness (#503)	3657 (10.0)	8.3	9.5	9.3	13.7
Anxiety (#212)	2767 (6.8)	4.7	6.6	7.4	9.3
Suicide ideation (#824)	2758 (7.5)	6.0	6.9	7.4	10.8
Suicide plan (#751)	2927 (8.1)	6.7	7.7	8.1	11.1
Suicide attempt (#199)	3184 (8.6)	5.9	8.8	8.7	11.7
Confounding Factors					
Peer support (#654)	12,906 (40.2)	40.9	39.5	39.2	41.7
Parental support (#1411)					
Low (0–1)	17,709 (47.2)	44.8	48.3	48.7	46.3
Medium (2)	9428 (27.7)	28.1	26.9	25.3	31.7
High (3–4)	7625 (25.1)	27.1	24.8	31.7	22.0
Body mass index (#3325)					
Normal	25,743 (77.3)	75.5	78.7	76.4	77.4
Under	3125 (9.6)	9.6	9.2	9.6	10.7
Overweight or obese	3980 (13.1)	14.9	12.1	14.0	12.0

**Table 3 ijerph-17-00132-t003:** Associations between soft drink use frequency and being attacked, in a physical fight, injury, being bullied, and truancy.

Carbonated Soft Drink Consumption	AOR (95% CI) ^1^	AOR (95% CI) ^2^
	Attacked	Attacked
Did not drink/past 30 days	1 (Reference)	1 (Reference)
<1 time/day	1.43 (1.28, 1.59) ***	1.41 (1.27, 1.57) ***
1 time/day	1.59 (1.42, 1.78) ***	1.53 (1.35, 1.74) ***
≥2 times/day	1.47 (1.22, 1.76) ***	1.51 (1.26, 1.82) ***
	In physical fight	In physical fight
Did not drink/past 30 days	1 (Reference)	1 (Reference)
<1 time/day	1.48 (1.31, 1.68) ***	1.47 (1.30, 1.66) ***
1 time/day	1.87 (1.62, 2.16) ***	1.87 (1.60, 2.17) ***
≥2 times/day	2.11 (1.81, 2.45) ***	2.20 (1.88, 2.56) ***
	Injury	Injury
Did not drink/past 30 days	1 (Reference)	1 (Reference)
<1 time/day	1.36 (1.18, 1.55) ***	1.35 (1.18, 1.54) ***
1 time/day	1.72 (1.51, 1.96) ***	1.72 (1.50, 1.96) ***
≥2 times/day	1.82 (1.52, 2.17) ***	1.94 (1.65, 2.29) ***
	Bullied	Bullied
Did not drink/past 30 days	1 (Reference)	1 (Reference)
<1 time/day	1.31 (1.17, 1.47) ***	1.34 (1.20, 1.50) ***
1 time/day	1.51 (1.34, 1.69) ***	1.48 (1.30, 1.68) ***
≥2 times/day	1.49 (1.27, 1.75) ***	1.51 (1.28, 1.78) ***
	Truancy	Truancy
Did not drink/past 30 days	1 (Reference)	1 (Reference)
<1 time/day	1.20 (1.07, 1.36) **	1.17 (1.02, 1.32) *
1 time/day	1.41 (1.23, 1.62) ***	1.42 (1.23, 1.65) ***
≥2 times/day	2.18 (1.84, 2.59) ***	2.27 (1.93, 2.68) ***

AOR = Adjusted Odds Ratio; CI = Confidence Interval; ^1^ adjusted for country only; ^2^ adjusted for country, sex, age, hunger, body weight status, peer support and parental support; *******
*p* < 0.001, ** *p* < 0.01, *****
*p* < 0.05.

**Table 4 ijerph-17-00132-t004:** Associations between soft drink use frequency and substance use.

Carbonated Soft Drink Consumption	OR (95% CI) ^1^	AOR (95% CI) ^2^
	Current tobacco use	Current tobacco use
Did not drink/past 30 days	1 (Reference)	1 (Reference)
<1 time/day	1.51 (1.24, 1.85) ***	1.44 (1.16, 1.17) ***
1 time/day	1.58 (1.28, 1.95) ***	1.63 (1.31, 2.03) ***
≥2 times/day	2.40 (1.83, 3.15) ***	2.68 (2.03, 3.53) ***
	Current alcohol use	Current alcohol use
Did not drink/past 30 days	1 (Reference)	1 (Reference)
<1 time/day	1.53 (1.29, 1.82) ***	1.62 (1.38, 1.90) ***
1 time/day	1.81 (1.46, 2.23) ***	2.09 (1.77, 2.48) ***
≥2 times/day	2.49 (1.96, 3.17) ***	2.90 (2.40, 3.50) ***
	**Drunkenness**	**Drunkenness**
Did not drink/past 30 days	1 (Reference)	1 (Reference)
<1 time/day	1.41 (1.15, 1.72) ***	1.54 (1.32, 1.80) ***
1 time/day	1.60 (1.25, 2.05) ***	1.91 (1.59, 2.29) ***
≥2 times/day	2.07 (1.61, 2.66) ***	2.56 (2.11, 3.11) ***
	**Cannabis use**	**Cannabis use**
Did not drink/past 30 days	1 (Reference)	1 (Reference)
<1 time/day	0.92 (0.66, 1.28)	1.05 (0.69, 1.59)
1 time/day	1.08 (0.71, 1.62)	1.41 (0.94, 2.13)
≥2 times/day	1.38 (0.88, 2.14)	1.97 (1.36, 2.87) ***
	**Amphetamine use**	**Amphetamine use**
Did not drink/past 30 days	1 (Reference)	1 (Reference)
<1 time/day	0.97 (0.66, 1.42)	1.13 (0.73, 1.76)
1 time/day	1.07 (0.67, 1.71)	1.28 (0.74, 2.22)
≥2 times/day	1.42 (0.83, 2.44)	2.14 (1.27, 3.62) **

AOR = Adjusted Odds Ratio; CI = Confidence Interval; ^1^ adjusted for country only; ^2^ adjusted for country, sex, age, hunger, body weight status, peer support and parental support; *** *p* < 0.001, ** *p* < 0.01, * *p* < 0.05.

**Table 5 ijerph-17-00132-t005:** Associations between soft drink use frequency and poor mental health.

Carbonated Soft Drinks Consumption	OR (95% CI) ^1^	AOR (95% CI) ^2^
	Loneliness	Loneliness
Did not drink/past 30 days	1 (Reference)	1 (Reference)
<1 time/day	0.87 (0.74, 1.01)	0.90 (0.76, 1.04)
1 time/day	0.92 (0.77, 1.09)	0.99 (0.81, 1.20)
≥2 times/day	1.34 (1.07, 1.66) **	1.38 (1.09, 1.72) **
	Anxiety	Anxiety
Did not drink/past 30 days	1 (Reference)	1 (Reference)
<1 time/day	1.15 (0.98, 1.36)	1.18 (0.99, 1.40)
1 time/day	1.36 (1.12, 1.64) **	1.47 (1.20, 1.81) ***
≥2 times/day	1.73 (1.43, 2.10) ***	1.88 (1.53, 2.32) ***
	Suicide ideation	Suicide ideation
Did not drink/past 30 days	1 (Reference)	1 (Reference)
<1 time/day	0.93 (0.80, 1.09)	0.94 (0.78, 1.12)
1 time/day	1.03 (0.86, 1.22)	1.14 (0.95, 1.37)
≥2 times/day	1.53 (1.24, 1.87) ***	1.63 (1.32, 2.02) ***
	Suicide plan	Suicide plan
Did not drink/past 30 days	1 (Reference)	1 (Reference)
<1 time/day	0.95 (0.82, 1.10)	0.95 (0.80, 1.13)
1 time/day	1.01 (0.85, 1.20)	1.08 (0.90, 1.31)
≥2 times/day	1.34 (1.11, 1.62) **	1.41 (1.15, 1.74) ***
	Suicide attempt	Suicide attempt
Did not drink/past 30 days	1 (Reference)	1 (Reference)
<1 time/day	1.01 (0.80, 1.28)	1.05 (0.84, 1.33)
1 time/day	1.07 (0.80, 1.42)	1.21 (0.94, 1.54)
≥2 times/day	1.37 (1.04, 1.82) *	1.69 (1.33, 2.16) ***

AOR = Adjusted Odds Ratio; CI = Confidence Interval; ^1^ adjusted for country only; ^2^ adjusted for country, sex, age, hunger, body weight status, peer support, and parental support; *** *p* < 0.001, ** *p* < 0.01, * *p* < 0.05.

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
