# Peer review of "High Carbonated Soft Drink Intake is Associated with Health Risk Behavior and Poor Mental Health among School-Going Adolescents in Six Southeast Asian Countries"

_ijerph, 2019, doi:10.3390/ijerph17010132_

Round 1

Reviewer 1 Report

Very nicely done! Paper will be of interest to the readership.

The only revision i recommend at this point is to go through the manuscript and wherever the word"subject" or "subjects" is used to reference the study participants that the word "subjects" be replaced with "participants" or "study participants" as appropriate. For example, line 17 in the abstract the word "subjects" should be replaced with "study participants". 

Author Response

the word "subjects" is replaced by study participants

Reviewer 2 Report

The manuscript provides very interesting information, and I believe the results and conclusions will be very useful for researchers in this topic. I really appreciate  the authors  have followed the suggestions made and I consider that the manuscript is ready to be published in this journal.

Author Response

no comments to respond to

This manuscript is a resubmission of an earlier submission. The following is a list of the peer review reports and author responses from that submission.

Round 1

Reviewer 1 Report

Pengpid and Peltzer: [IJERPH] Manuscript ID: ijerph-581683

Abstract: Abstract is pretty well constructed but parts could be cleaned up.  It presented need for study, and an outline of study results.

These sentences were rough: “In the final adjusted logistic regression model, any and once and ≥two times CSD intake per day was associated with attacked, injury, physical fight, bullied, school truancy, tobacco use, alcohol use and lifetime drunkenness.” Perhaps reorganize to smaller sentences?

“Results suggest that CSD intake is associated with several externalizing and internalizing health compromising behaviours” Suggest reorganizing: CSD intake in developing countries is associated with several externalizing and internalizing health compromising behaviors that are similar to those observed in the USA (…do you wish to use your term high-income counties?)

Introduction: It covered the main points, but seemed a bit brief.  Perhaps consider greater discussion of what internal and external behaviors really are?

Last sent Para1 rough- avoid using same word more than once in a sentence (and)

Consider converting Para2 (one long run-on sentence) into two or even three more developed sentences.

Para3: Avoid quotations, please paraphrase

Para4: Consider explaining why your work is important?  For example economic and social costs of suicide, bullying, etc or something of the like?

Methods: Please avoid quotations, please add additional details regarding delivery.  Reader may not want to read Ref #11 to know how study was administered.

Table 1 needs a better more detailed legend/description, although I like the content.

Results:  A lot of your data was best presented (as you nicely did) in table format.  One suggestion would be to include the % of study respondents who originated in each of the countries, was this a balanced design or skewed to mostly one or two countries with minor contributions from the others?  You may also wish to consider including the sex ration (male vs female) as well as the average (standard deviation) for ages and body mass index.  These two variables might be included in your statistical model to predict where the CSD-internal/external behavior association was greatest.  For example were older or younger respondents more effected?

Discussion: Could be improved by expanding statistical analysis (see above) and better editing/rewriting what you have.

Para1: cut this long one-sentence paragraph into two perhaps three more specific sentences.

Para2: last sentence is rough

Para4: last sentence uses larger non-uniform font size

Conclusion: Seems appropriate

Acknowledgements: Avoid quotations

References: needs uni

Author Response

Reviewer 1:
Abstract: Abstract is pretty well constructed but parts could be cleaned up. It presented need for study, and an outline of study results.
These sentences were rough: “In the final adjusted logistic regression model, any and once and ≥two times CSD intake per day was associated with attacked, injury, physical fight, bullied, school truancy, tobacco use, alcohol use and lifetime drunkenness.” Perhaps reorganize to smaller sentences?
Response: changed to below
In the final adjusted logistic regression model CSD intake was associated with attacked, injury, physical fight, bullied, school truancy, tobacco use, alcohol use and lifetime drunkenness.
“Results suggest that CSD intake is associated with several externalizing and internalizing health compromising behaviours” Suggest reorganizing: CSD intake in developing countries is associated with several externalizing and internalizing health compromising behaviors that are similar to those observed in the USA (…do you wish to use your term high-income counties?)
Introduction: It covered the main points, but seemed a bit brief. Perhaps consider greater discussion of what internal and external behaviors really are?
Response: changed accordingly to below
CSD intake in developing countries is associated with several health risk behaviours and poor mental health that are similar to those observed in high-income countries.

Last sent Para1 rough- avoid using same word more than once in a sentence (and)
Response: Corrected, as below
Several local or national and one multi-country study in high-income countries, including one study in China and one in Malaysia, found associations between CSD intake and health compromising behaviours.
Consider converting Para2 (one long run-on sentence) into two or even three more developed sentences.
Response: Corrected
Para3: Avoid quotations, please paraphrase
Response: Corrected
Para4: Consider explaining why your work is important? For example economic and social costs of suicide, bullying, etc or something of the like?
Response: below is added
A balanced diet is vital during the developmental stage of adolescence for physical and mental development, and therefore it is of public health concern to understand the relationship between CSD intake and health risk behaviour and poor mental health [3].
Methods: Please avoid quotations, please add additional details regarding delivery. Reader may not want to read Ref #11 to know how study was administered.
Response: Corrected to below
The GSHS uses a cluster sampling design in two stages (schools were selected by probability to size sampling and random selection of class rooms with students 13 to 15 years old) in order to produce nationally representative samples of school children in middle schools in each study country [11]. All students attending a selected class were eligible to participate, regardless of their age, and completed a self-administered questionnaire in their language under the supervision of trained external survey administrators [11]. Ethics review boards in each country approved the GSHS, and informed consent was obtained from the students, parents and/or school authorities [11].

Table 1 needs a better more detailed legend/description, although I like the content.
Response: changed to below
Table 1. Description of variables used in this survey.

Results: A lot of your data was best presented (as you nicely did) in table format. One suggestion would be to include the % of study respondents who originated in each of the countries, was this a balanced design or skewed to mostly one or two countries with minor contributions from the others?
Response: below is added
Among the six study countries, the lowest number of participants were from Bangladesh (N=2989, 8.3%) and the highest number from Indonesia (N=11142, 30.8%).
You may also wish to consider including the sex ration (male vs female) as well as the average (standard deviation) for ages and body mass index.
Response: below is added [age median is already reported, while reporting the mean BMI was not felt necessary]
The proportion of female students were 55.1% and male students 44.9%, and 6.6% had mostly or always experienced hunger in the past month. About one in four of the students (26.9%) had been in a physical fight in the past 12 months, 38.6% had been attacked, 38.1% had sustained a serious injury, 29.5% had been bullied, and 26.3% “missed classes or school without permission” in the past month. Regarding substance use, 13.2% of participants were current tobacco users, 10.5% current alcohol users, 10.2% had ever been drunk, 3.6% had ever used cannabis and 2.8% had ever used amphetamine. In terms of mental health, 10.0% of students reported loneliness, 6.8% anxiety, 7.5% suicide ideation, 8.1% had a suicide plan and 8.6% had attempted suicide in the past 12 months. Two in five students (40.2%) received almost always or always peer support in the past month, 52.8% had medium or high parental support, 9.6% were underweight and 13.1% overweight or obese.
These two variables might be included in your statistical model to predict where the CSD-internal/external behavior association was greatest. For example were older or younger respondents more effected?
Response: age and sex are included in the models, but not separately reported
Discussion: Could be improved by expanding statistical analysis (see above)
Response: The objective was not to report on covariates, other than soft drink consumption, which could be subject to a separate paper
and better editing/rewriting what you have.
Response: Corrected
Para1: cut this long one-sentence paragraph into two perhaps three more specific sentences.
Response: corrected as below
The study found a prevalence of no past-month CSD intake of 23.9%, 38.8% <once/day, 19.9% once a day and 17.5% ≥two times/day in Bangladesh, Indonesia, Laos, Philippines, Thailand and Timor-Leste in 2014-2015, which is an increase to a study in Southeast Asia (Malaysia, Maldives) in 2009-2012, 25.8%, 42.1%, 16.2% and 15.4%, respectively [12]. However, the prevalence of ≥once CSD daily intake (37.8%) in this study across six Southeast Asian countries was lower than the corresponding prevalence in 53 developing countries (54.3%) [12].

Para2: last sentence is rough
Response: Corrected, as below
This result underlines the importance of tackling a clustering of CSD intake with a number of health risk behaviours and poor mental health in health promotion programmes in this school population.

Para4: last sentence uses larger non-uniform font size
Response: Corrected
Conclusion: Seems appropriate
Acknowledgements: Avoid quotations
Response: Corrected
References: needs uni
Response: Corrected

Reviewer 2 Report

Hello

This is a very exciting and interesting study. Unfortunately, I cannot recommend publication in the present format. 

There is much too much data for a single paper - should be at least two papers. The tables are not clear the way they are and need to be divided into more tables with clear headers - not using footnotes to label columns. A much longer methods section is required with detailed explanation of the survey and how access is provided. It was only through reading a footnote that I became aware that this was a WHO survey and that there was broad access to the study. The methods need to provide details of the subjects and the statistics A quote with a reference to the WHO study website is insufficient. There is not mention of your institutional human subjects review panel - did they review your study? If review was waved because the study data was acquired from WHO and considered "non-human" subject data.. then this needs  to be stated. The results need to be described in the text - not assume that the reader can look at the tables and determine what they mean. The discussion is much too short and does not adequately "interpret" the results - but rather states the results. Again this needs to be turned into several papers not one so the manuscript can adequately present and discuss the data.

Author Response

This is a very exciting and interesting study. Unfortunately, I cannot recommend publication in the present format.
There is much too much data for a single paper - should be at least two papers.
Response: this paper merely looks at various outcomes of soft drink consumption, as has been done previously with European countries, other papers look at individual health risk behaviours as outcomes
The tables are not clear the way they are and need to be divided into more tables with clear headers - not using footnotes to label columns.
Response: The table descriptions are expanded, we do not see the need to make more tables and we cannot see how “footnotes are used to label columns”
A much longer methods section is required with detailed explanation of the survey and how access is provided. It was only through reading a footnote that I became aware that this was a WHO survey and that there was broad access to the study. The methods need to provide details of the subjects and the statistics A quote with a reference to the WHO study website is insufficient.
Response: below is added
Details about the GSHS and its data can be publicly accessed [11]. The GSHS uses a cluster sampling design in two stages (schools were selected by probability to size sampling and random selection of class rooms with students 13 to 15 years old) in order to produce nationally representative samples of school children in middle schools in each study country [11]. All students attending a selected class were eligible to participate, regardless of their age, and completed a self-administered questionnaire in their language under the supervision of trained external survey administrators [11]. Ethics review boards in each country approved the GSHS, and informed consent was obtained from the students, parents and/or school authorities [11].
There is not mention of your institutional human subjects review panel - did they review your study? If review was waved because the study data was acquired from WHO and considered "non-human" subject data.. then this needs to be stated.
Response: As this was a secondary data analysis, this was not required
The results need to be described in the text - not assume that the reader can look at the tables and determine what they mean.
Response: below is added
The proportion of female students were 55.1% and male students 44.9%, and 6.6% had mostly or always experienced hunger in the past month. About one in four of the students (26.9%) had been in a physical fight in the past 12 months, 38.6% had been attacked, 38.1% had sustained a serious injury, 29.5% had been bullied, and 26.3% “missed classes or school without permission” in the past month. Regarding substance use, 13.2% of participants were current tobacco users, 10.5% current alcohol users, 10.2% had ever been drunk, 3.6% had ever used cannabis and 2.8% had ever used amphetamine. In terms of mental health, 10.0% of students reported loneliness, 6.8% anxiety, 7.5% suicide ideation, 8.1% had a suicide plan and 8.6% had attempted suicide in the past 12 months. Two in five students (40.2%) received almost always or always peer support in the past month, 52.8% had medium or high parental support, 9.6% were underweight and 13.1% overweight or obese.

The discussion is much too short and does not adequately "interpret" the results - but rather states the results. Again this needs to be turned into several papers not one so the manuscript can adequately present and discuss the data.
Response: This is for a brief report adequately discussed.

Reviewer 3 Report

This is a novel and original research which open a new area of study in relationship between carbonated soft drink intake and various risk behaviours in adolescents in high-income countries. Results showed that carbonated soft drink intake is associated with several externalizing and internalizing health compromising behaviours. The sample was enough to analyze 6 countries and showed similar results as other studies analyzed in previously researches. The rational of this study is clear. Methodological issues are clear. However, I offer the following suggestions for improvement

Improve participants description

Explain training application of the results obtained for school population

Review the format of the manuscript. There are some different letter sizes along the document.

Author Response

This is a novel and original research which open a new area of study in relationship between carbonated soft drink intake and various risk behaviours in adolescents in high-income countries. Results showed that carbonated soft drink intake is associated with several externalizing and internalizing health compromising behaviours. The sample was enough to analyze 6 countries and showed similar results as other studies analyzed in previously researches. The rational of this study is clear. Methodological issues are clear. However, I offer the following suggestions for improvement
Improve participants description
Response: added as below
Among the six study countries, the lowest number of participants was from Bangladesh (N=2989, 8.3%) and the highest number from Indonesia (N=11142, 30.8%). The proportion of female students were 55.1% and male students 44.9%, and 6.6% had mostly or always experienced hunger in the past month. About one in four of the students (26.9%) had been in a physical fight in the past 12 months, 38.6% had been attacked, 38.1% had sustained a serious injury, 29.5% had been bullied, and 26.3% “missed classes or school without permission” in the past month. Regarding substance use, 13.2% of participants were current tobacco users, 10.5% current alcohol users, 10.2% had ever been drunk, 3.6% had ever used cannabis and 2.8% had ever used amphetamine. In terms of mental health, 10.0% of students reported loneliness, 6.8% anxiety, 7.5% suicide ideation, 8.1% had a suicide plan and 8.6% had attempted suicide in the past 12 months. Two in five students (40.2%) received almost always or always peer support in the past month, 52.8% had medium or high parental support, 9.6% were underweight and 13.1% overweight or obese
Explain training application of the results obtained for school population
Response: below is stated
Frequent CSD intake and its associated health risk behaviours and poor mental health should be targeted in school health promotion interventions.

Review the format of the manuscript. There are some different letter sizes along the document.
Response: Corrected

Round 2

Reviewer 1 Report

Pengpid and Peltzer: [IJERPH] Manuscript ID: ijerph-581683

While the authors did work very hard to edit and revise their manuscript I still thought it was a bit rough for an MDPI publication.  While t the topic is important I thought the authors just need more content/depth to how they approached it in the intro, methods, results and discussion.  Sorry to have to say this. I guess major edits means really taking time to rethink and revise.  Authors should probably have taken more time to truly ponder how to improve what they had submitted prior to re-submitting.

Abstract:  Improved but still rough in spots and could be better. Abstract determines if reader will move to the rest of the paper and probably needs to be cleanest reading portion of a good manuscript.   Line 4 would be better is written to only use “and” once.  Line 8: , the prevalence of no CSD use…this sound better? Still rough: …associated with attacked, injury,….  ……CSD was associated with a history of loneliness, anxiety….

Introduction: Still seemed a bit short/brief/rough unfortunately.

PP1sentance 3 needs work

PP2Line6: In a study OF XXXX in 26 high-income countries…

PP3: still rough

Methods:

PP1: Better but you could improve it by reducing the number of times you cite ref #11.  Did respondents complete survey using paper or computer?  This could be indicative of affluence of students used in survey.  Seems that three short paragraphs is not enough.

Table 1 description is still a bit brief, for example:   Table 1. Global School-based Student Health Survey variables, questions, and response options evaluated in this study of 20140-2015 respondents.

Typo/rough:  The GSHS measure as used in this survey is shown in Table 1.  Consider: The survey parameters of the GSHS are described in detail in Table 1 [11].

Results:  Paragraph #1 is much improved, thank you. 

Paragraph #2, still have “and” three times in same sentence, still very rough sorry to say.

Tables 3,4 and 5 would be easier to understand if you could insert Adjusted by country only and details2

Directly into the column at top of table…..going to foot note at bottom seemed confusing.

Discussion:

Paragraph #1 is still rough, as are last sentence of paragraph 2 and first sentence of paragraph 3, though I do like paragraph 4.   The study limitations only consists of one sentence?  This seems to have over looked many possible problems, did the authors accidently delete more content?

Author Response

Abstract: Improved but still rough in spots and could be better. Abstract determines if reader will move to the rest of the paper and probably needs to be cleanest reading portion of a good manuscript. Line 4 would be better is written to only use “and” once. Line 8: , the prevalence of no CSD use…this sound better? Still rough: …associated with attacked, injury,…. ……CSD was associated with a history of loneliness, anxiety….
Response: Corrected
Introduction: Still seemed a bit short/brief/rough unfortunately.
PP1sentance 3 needs work
Response: Corrected
PP2Line6: In a study OF XXXX in 26 high-income countries…
Response: Corrected
PP3: still rough
Response: Corrected
Methods:
PP1: Better but you could improve it by reducing the number of times you cite ref #11. Did respondents complete survey using paper or computer? This could be indicative of affluence of students used in survey. Seems that three short paragraphs is not enough.
Response: below is added
All students attending a selected class were eligible to participate, regardless of their age, and completed a self-administered questionnaire on a computer scanable answer sheet in their language under the supervision of trained external survey administrators
Table 1 description is still a bit brief, for example: Table 1. Global School-based Student Health Survey variables, questions, and response options evaluated in this study of 20140-2015 respondents.
Response: Corrected
Typo/rough: The GSHS measure as used in this survey is shown in Table 1. Consider: The survey parameters of the GSHS are described in detail in Table 1 [11].
Response: Corrected accordingly
Results: Paragraph #1 is much improved, thank you.
Paragraph #2, still have “and” three times in same sentence, still very rough sorry to say.
Response: Corrected
Tables 3,4 and 5 would be easier to understand if you could insert Adjusted by country only and details2
Response: added
Directly into the column at top of table…..going to foot note at bottom seemed confusing.
Response: We believe it is better, as it is, at the bottom of the Table
Discussion:
Paragraph #1 is still rough, as are last sentence of paragraph 2 and first sentence of paragraph 3, though I do like paragraph
R: Corrected
4. The study limitations only consists of one sentence? This seems to have over looked many possible problems, did the authors accidently delete more content?
Response: more is added, as in below
The study was limited by its cross-sectional design of a school population and measures using self-report. Several concepts in this study were assessed with single items, and future investigations may consider including more comprehensive measures. The study did also not assess dietary intake in terms of foods that are high in refined sugar content, such as candy and chocolate, which could be important in relation to sweetened beverage consumption.